# An Individuality of Response to Cannabinoids: Challenges in Safety and Efficacy of Cannabis Products

**DOI:** 10.3390/molecules28062791

**Published:** 2023-03-20

**Authors:** Sarunya Kitdumrongthum, Dunyaporn Trachootham

**Affiliations:** Institute of Nutrition, Mahidol University, Nakhon Pathom 73000, Thailand

**Keywords:** cannabis, toxicity, individual variation, response, genetic polymorphism, hormesis, functional food

## Abstract

Since legalization, cannabis/marijuana has been gaining considerable attention as a functional ingredient in food. ∆-9 tetrahydrocannabinol (THC), cannabidiol (CBD), and other cannabinoids are key bioactive compounds with health benefits. The oral consumption of cannabis transports much less hazardous chemicals than smoking. Nevertheless, the response to cannabis is biphasically dose-dependent (hormesis; a low-dose stimulation and a high-dose inhibition) with wide individuality in responses. Thus, the exact same dose and preparation of cannabis may be beneficial for some but toxic to others. The purpose of this review is to highlight the concept of individual variations in response to cannabinoids, which leads to the challenge of establishing standard safe doses of cannabis products for the general population. The mechanisms of actions, acute and chronic toxicities, and factors affecting responses to cannabis products are updated. Based on the literature review, we found that the response to cannabis products depends on exposure factors (delivery route, duration, frequency, and interactions with food and drugs), individual factors (age, sex), and susceptibility factors (genetic polymorphisms of cannabinoid receptor gene, N-acylethanolamine-hydrolyzing enzymes, THC-metabolizing enzymes, and epigenetic regulations). Owing to the individuality of responses, the safest way to use cannabis-containing food products is to start low, go slow, and stay low.

## 1. Introduction

Cannabis or marijuana use is becoming increasingly popular around the world, especially after the legalization of cannabis products for medical, dietary, and recreational purposes in several countries such as Canada, Georgia, Malta, Mexico, South Africa, Thailand, Uruguay, the United States (in 19 states, two territories, and the District of Columbia), and the Capital Territory in Australia [1]. Some countries such as Germany also plan to legalize cannabis products [1]. The World Health Organization (WHO) reported that cannabis is the world’s most popular recreational drug, with 147 million people consuming it (2.5% of the world population) [2]. In 2022, the legal sales of adult-used cannabis worldwide amounted to USD 22 billion, and this is expected to reach over USD 33 billion by 2025 [3]. Cannabis is a common term for numerous products of the plant *Cannabis sativa*. 

The key psychoactive component in cannabis is ∆-9 tetrahydrocannabinol (THC). Cannabinoids are compounds with structures that resemble THC [2]. THC and cannabinoids can alter in terms of physiologic function, especially regarding the nervous system. Cannabis products have been used effectively for alleviating neuropathic pain, chemotherapy-induced nausea and vomiting, and muscle spasticity symptoms [4]. However, toxicities associated with cannabis use are commonly found. Acute intoxication can cause euphoria, perception alterations such as time and spatial distortion, the intensification of ordinary sensory experiences, and motor impairment [5]. Furthermore, some users may experience adverse psychological reactions such as panic, fear, or depression [5]. 

Since the legalization of cannabis products, more food products containing THC and cannabinoids have been made widely available and easily accessible [6]. Moreover, synthetic cannabinoids, which can cause more severe toxicities than natural phytocannabinoids, are increasingly being generated and used in those products [6]. Such changes can increase the chances of exposure to sensitive groups, especially children [6]. 

The most unique characteristic of cannabis use is the large variety in individual responses to cannabis products. With the same dose and preparation of cannabis products, some people may benefit from the use, while some may face unwanted side effects that require medical attention [7]. Furthermore, in defining the safe dose of cannabis products, focusing on a specific active compound such as THC or cannabidiol (CBD) is insufficient for predicting the effects. Due to the differential profiles of the 140 other phytocannabinoids, a wide range of effects can be observed [8]. Such variation leads to challenges in setting health-based guidance values and standardized cannabis units for safety control [9,10].

Cannabis has been used as a medicine for several years. The application of cannabis as a substitute for prescription drugs has included its use as a pain medication, antidepressant, antipsychotic, anti-epileptic, arthritis medication, and others (for sleep, anxiety, and ADHD, and as a muscle relaxant) [11,12,13]. Interestingly, a recent survey in Denmark with 1546 respondents revealed that 65.8% of people reported cannabis as a “much more effective” treatment of their condition(s) than prescription drugs [11]. Additionally, the majority (85.5%) indicated that the side effects of prescription drugs were “much worse” than the side effects of cannabis [11]. The list of used cannabis-related drugs, their therapeutic uses, their chemistry, and the pharmacology data have been extensively summarized in several recent review articles [11,12,13]. Nevertheless, literature reviews on the individuality of responses to cannabinoids are still lacking. 

The purpose of this study is to highlight the concept of individual variation in response to cannabinoids, which leads to the challenge of establishing standard safe doses of cannabis products for the general population. In this review, the mechanisms of action and acute and chronic toxicities are updated. Importantly, the individuality of responses to cannabis products, and factors contributing to the large variation in responses, are discussed. Future research questions and directions are addressed. 

This review sheds light on potential evidence-based approaches to the safer use of cannabis-containing food products, with the consideration of individual variations in cannabinoid responses. The response to cannabis products depends on exposure factors, individual factors, and susceptibility factors. Thus, the safest use of cannabis-containing food products is to start low, go slow, and stay low. The establishment of a safe dose of cannabinoids in cannabis-containing food products for regulatory purposes should consider toxicological data, the factors affecting the individuality of responses, the delivery system, and the pharmacokinetic interactions with other food. 

## 2. The Active Substances in Cannabis, the Endocannabinoid System (ECS), and Metabolism

### 2.1. The Active Substances in Cannabis

*Cannabis sativa* L. (*C. sativa*), generally known as cannabis, hemp, or marijuana, is a member of the *Cannabaceae* family. The genus *Cannabis* is subdivided into three species, *Cannabis sativa*, *Cannabis indica*, and *Cannabis ruderalis* [14,15,16,17]. Cannabis plants are thought to have originated in Central and South Asia, including India, Pakistan, Afghanistan, China, Kazakhstan, and Uzbekistan. Their leaves and female flowers contain various secondary metabolites, whereas the seeds are a source of omega-3 fatty acids [18]. 

The cannabis plant is abundant in phytochemicals, with over 545 recognized substances and over 140 types of cannabinoids. The major cannabinoids found in cannabis are C21 terpene phenolic compounds and C22 in the carboxylate form. In addition, the plant also includes a variety of noncannabinoid terpenes and phenols, alkanes, sugars, nitrogenous compounds (e.g., spermidine alkaloids or muscarine), flavonoids, noncannabinoid phenols, phenylpropanoids, steroids, and fatty acids [15,16,19,20]. The structures of the leading bioactive compounds in *C. sativa* were revealed in a study by Liu and colleagues [21]. 

Among the phytocannabinoids, the compounds with the most potent physiological and psychotogenic effects include trans- delta-9-tetrahydrocannabinol (THC) and cannabidiol (CBD), alongside others such as cannabinol (CBN), cannabigerol, and cannabichromene [22,23]. THC takes the form of four stereoisomers, while only the (–)-trans-isomer occurs in nature. Two structurally similar compounds, ∆9-tetrahydrocannabinol-2-oic acid and ∆9-tetrahydrocannabinol-4-oic acid (THCA), are also detected. THCA can be partially converted to THC by heat. The other active isomer of THC is delta-8-Tetrahydrocannabinol (∆8-THC), which is present at lower amounts [16,24]. 

THC is the key and most potent psychoactive ingredient in cannabis, and that which causes highness upon intake. In addition, THC provides antiemetic, anti-inflammatory, and pain-relieving effects for neuropathy and chronic pain. Therefore, the level of THC in cannabis products can determine their potencies [15]. Depending on the species, the amount of the psychoactive component THC can vary. Due to its higher content of THC content than other species, *C. sativa* is the most commonly explored. 

The categorization of cannabis as either drug-type or nondrug-type is based on THC concentration [25]. Drug-type cannabis with a psychotropic effect (marijuana) contains 1.0–20% THC; the intermediate type contains 0.3–1.0% THC; and the fiber type (hemp), which is found in textiles and food, contains 0.3% THC [26]. *Cannabis sativa* L. comprises all forms of hemp and marijuana, with high genomic and phenotypic variation across multiple lineages [27]. Marijuana lineages are utilized for medical and recreational consumption, while hemp lineages are exploited in the industry for fiber or oil extraction. 

The genetic background and environmental conditions of cultivation influence the variations in the chemical components and properties of cannabis [27,28]. A recent study in the US reported the chemotaxonomic analysis of various cannabis samples grown in six states of the US [29]. The results show that there is variation in the bioactive compounds, as the majority of samples were THC-dominant, while the minority showed a balance of THC: CBD and CBD dominance. Though the total amount of terpenes positively correlated with the total amount of cannabinoids, there is variation in the diversity of type. THC-dominant (“Type I”) cultivars display a more diverse array of terpene profiles than those of the balanced THC: CBD (“Type II”) and CBD-dominant (“Type III”) cultivars [29].

### 2.2. The Endocannabinoid System (ECS)

The biological effects of phytocannabinoids from the cannabis plant are mediated by cannabinoid receptors (CNR), members of the G-protein-coupled receptor family. Two types of CNR have been identified, including cannabinoid receptor 1 (CB1), which is found mainly in the central nervous system, and cannabinoid receptor 2 (CB2), which is found in peripheral tissues such as immune cells. Both receptors bind to endocannabinoids, leading to signal transduction and downstream effects [30]. The classical endocannabinoids include *N*-arachidonoylethanolamine (anandamide, AEA) and 2-arachidonoylglycerol (2-AG) [31]. 

AEA is biosynthesized from membrane-phospholipid precursors via the action of *N*-acyltransferase (NAT) and N-acyl-phosphatidylethanolamine-specific phospholipase D (NAPE-PLD) enzymes. The generation of 2-AG is catalyzed by the diacylglycerol lipases (DAGLα/β) enzyme. The endocannabinoid system’s activity can be terminated by hydrolysis and/or oxidation. AEA is hydrolyzed to arachidonic acid and ethanolamine by fatty acid amide hydrolase (FAAH), while 2-AG is hydrolyzed to arachidonic acid and glycerol by monoacyl-glycerol lipase (MAGL) [30,31,32]. AEA and 2-AG can be oxidized by cyclooxygenase-2 and numerous lipoxygenases [33]. 

More molecules have recently been identified as endogenous cannabinoids, such as 2-arachidonoyl glyceryl ether (noladin ether, 2-AGE), O-arachidonoylethanolamine (virodhamine), *N*-arachidonoyldopamine (NADA), and oleic acid amide (oleamide, OA) [34]. AEA has a high affinity to the CB1 receptor as a partial agonist, and a low affinity to the CB2 receptor. In contrast, 2-AG shows a moderate affinity for both receptors and acts as a complete agonist [35]. The exogenous THC is a partial CB1 and CB2 agonist, whereas the affinity of phytocannabinoid CBD to both CB1 and CB2 receptors is low [36,37]. 

Recently, new types of phytocannabinoids have been identified and extracted from *C. Sativa*, including ∆9-tetrahydrocannabiphorol (∆9-THCP) and cannabidiphorol (CBDP), which are the homologs of ∆9-THC and CBD, respectively. ∆9-THCP has a high affinity to both CB receptors, while the pharmacological effects of CBDP are unknown [15,38]. Binding between endocannabinoids and CB receptors results in the modulation of synaptic transmission in multiple pathways, regardless of synaptic nature and transmission duration [39]. 

Endocannabinoids have been shown to regulate a variety of different receptors and channels, including TRP channels and the G-protein-coupled receptors GPR55, GPR18, GPR119, γ-aminobutyric acid (GABA), and glycine receptors. THC and several phytocannabinoids mediate their bioactivities through the cannabinoid receptors, and further act as agonists to GPR55, GPR18, PPAR, transient TRPA1, TRPV2, TRPV3, and TRPV4. In contrast, endocannabinoids act as antagonists to TRPM8 and 5-HT3A [31,35,40].

### 2.3. Cannabinoid Receptors and Cannabinoids Effects

CB1 and CB2 are members of the G-protein-coupled receptor family, and are distributed throughout our bodies. In humans, CB1 is encoded by the *CNR1* gene and consists of 472 amino acids. CB1 receptors are mainly located on central and peripheral neurons [30]. Full-length CB1 is found in the brain and skeletal muscle. In contrast, the CB1b isoform is more abundant in liver and pancreatic islet cells [41]. CB1 expression is highest in the olfactory bulb, hippocampus, basal ganglia, and cerebellum. It expresses at a moderate level in the cerebral cortex, septum, amygdala, hypothalamus, parts of the brainstem, and the dorsal horn of the spinal cord [42]. In the peripheral nervous system (PNS), CB1 is highly expressed in sympathetic nerve terminals [43]. CB1 is also expressed in some non-neuronal cells, such as immune cells [37]. Interestingly, variants in the *CNR1* gene were found in heavy cannabis users [30,44]. 

CB2 is derived from the *CNR2* gene and composed of 360 amino acids. At the protein level, CB2 shares 44% sequence homology with CB1 [30]. CB2 receptors are found primarily in immune cells and some neurons both inside and outside the brain. Moreover, one CB2 isoform is mainly expressed in the testes, but present at lower levels in brain reward regions. Another CB2 isoform is highly expressed in the spleen, but has also been found at lower levels in the brain [37,45]. 

Owing to its more robust expression in multiple types of cells, CB2 has a significant advantage over CB1 as a therapeutic target. On the other hand, CB1 is predominantly expressed in the central nervous system (CNS), which is the primary receptor responsive to ∆9-THC and contributes to the psychoactive effects of cannabis [38].

### 2.4. THC Metabolism

The metabolism of THC occurs mainly in the liver and other organs, including the brain, small intestine, heart, and lungs [43,44]. Cytochrome P450s (CYPs), including CYP2C9, CYP2C19, and CYP3A4, are the key phase I enzymes responsible for the liver metabolism of THC [46,47].

There are over 100 THC metabolites, and most of them are monohydroxylated compounds [48]. The C-11 position in THC’s structure is the most attacked site of CYPs in humans, and the major metabolites of THC are the active metabolite 11-hydroxy-THC (11–OH–THC) and the inactive metabolite 11-carboxy-THC (THC–COOH) [48]. Subsequently, these metabolites undergo glucuronidation as a phase II reaction, or, less commonly, conjugation with amino acids, fatty acids, sulfate, and glutathione [48].

## 3. Acute Toxicity of Cannabis Use

Cannabis intoxication is proportional to the dose and duration of exposure, and its absorption depends on the delivery route and concentration. The most common method of cannabis consumption is through the airways. Smoking is common for recreational purposes, whereas vaporization is utilized for both recreational and medicinal purposes [46]. Compared to other methods of use, smoking has the highest potential for cannabis addiction due to the rapid and efficient transport of the drug to the brain [49,50]. Within a few seconds of inhalation, plasma concentrations of THC and CBD are detectable, and they reach a peak after 3 to 10 min [51]. Since THC is highly lipophilic, serum concentrations can peak in 15 to 30 min and last for up to 4 h [50]. However, the peak plasma concentration (Cmax) and the time to reach peak (Tmax) can vary according to the individual host metabolism, as well as the inhalation time, the number, interval, and duration of puffs, inhalation volume, particle size, and inhalation device [51,52].

The ingestion or oral administration of cannabis can be in the form of capsules, food, or cannabis-infused drinks. The benefit of this route is that the number of hazardous chemicals generated during oral consumption is much less than that during smoking [51,53]. Orally administered cannabis shows bioavailability decreased by around 5–20% compared to inhalation, because of chemical breakdown in gastric acid and significant first-pass metabolism in the liver [49]. Compared to inhalation, oral usage results in the delayed onset of effects, with THC peak concentrations often occurring within 1 to 2 h and lasting up to 8 h. However, the duration of psychoactive effects is still prolonged, with a delayed return to baseline ranging from 30 min to 3 h, and extending up to 12 h [50,54]. 

Another method of oral administration is oromucosal delivery, which is utilized for therapeutic purposes. Compared to the oral administration route, absorption through the buccal mucosa is faster, resulting in higher plasma concentrations [55]. By administering drugs sublingually, the first-pass hepatic metabolism is avoided [53]. There are currently no deaths reported due to direct or acute poisoning. However, comas have frequently occurred following unintended intake by children [56]. The acute toxicities associated with the inhalation and ingestion administration of cannabis products are as follows. 

### 3.1. Inhalation and Smoking

About 50% of the THC and other cannabinoids in cannabis are released into the smoke. Experienced users often inhale deeply and hold the smoke in their airways for a few seconds before exhaling it. Such a method causes all of the cannabinoids in the smoke to enter the bloodstream, thus retaining their effect [51,54]. Cannabinoids affect a person’s mood due to their euphoriant potential, which is the ability to produce a “high”, as well as the feelings of “pleasure” and “relaxation” [55]. The inhalation of 2 to 3 mg THC can impair attention, focus, short-term memory, and executive functioning in both adolescents and adult populations [48]. Adults who inhale THC at doses exceeding 7.5 mg/m^2^ could experience more severe symptoms, including hypotension, panic, anxiety, myoclonic jerking/hyperkinesis, delirium, respiratory depression, and ataxia [57]. Using cannabis as a medicine via the inhalation route can cause adverse effects, including somnolence, amnesia, coughing, nausea, dizziness, euphoric mood, hyperhidrosis, and paranoia [58].

### 3.2. Ingestion

The consumption of edible products is the most common form and route of exposure for the general population of all ages. Adults who intend to consume cannabis-containing food may experience adverse effects from overuse. In contrast, the pediatric population mostly consumes edible cannabis products unintentionally. The adverse effects of ingesting cannabis, especially following exposure to resins and liquid concentrates, include CNS sedation and respiratory distress requiring intubation [59]. When taken orally, doses of 5–20 mg of THC have the potential to impair short-term memory, induce a loss of attention and executive functioning, and impair memory in general. The ingestion of a highly concentrated THC-containing food product (i.e., 5–300 mg THC) is the most frequent cause of acute toxicity in young children [50]. In cases of unintentional cannabis ingestion in children, cannabis resin was the most commonly consumed product, followed by cookies, leftover smoking products such as hashish (dried cannabis bar) or joints (rolled cannabis cigarettes), medical cannabis, candies, and beverages [60]. 

The most common adverse clinical sign is lethargy, followed by ataxia. In addition, tachycardia, mydriasis, and hypotonia have frequently been noted. Less specific symptoms reported include nausea and vomiting, bradycardia, bradypnea, hypotension, and respiratory distress [48,58]. All cases were either hospitalized or admitted to the emergency department. Some were admitted to a pediatric intensive care unit and intubated [58,60].

Significant acute cannabis intoxication impairs memory and cognition, motor function, reaction speed, and psychomotor performance. Cannabis intoxication in the general population is characterized by a variety of symptoms, including those related to the cardiovascular system (tachycardia, hypertension), the ophthalmological system (conjunctival injection, nystagmus), the respiratory system (tachypnea, bradypnea), the gastrointestinal system (tachycardia, bradypnea) and the neurological system (dry mouth, increased appetite) [52,61]. In addition to euphoria, some symptoms such as anxiety, dysphoria, panic, paranoia, sleepiness, somnolence, slurred speech, and memory lapses have been reported. 

The noticeable physiological effects of cannabis usage are an elevation in blood pressure, a change in heart rate, and an effect on blood-oxygen saturation [56,59,62]. Among all adverse events associated with cannabis’ acute toxicity, neurotoxicity is most commonly found in populations of all ages. However, the symptoms are varied. While symptoms of CNS suppression such as lethargy and ataxia are mostly seen in children, symptoms of CNS excitation such as anxiety, paranoia, panic attacks, lightheadedness (a feeling of being close to fainting), dizziness, and vertigo are more common in adults [57]. Interestingly, equivalent incidences of CNS depression/excitation have been reported in adolescents [57]. 

## 4. Chronic Toxicity of Cannabis Use

The chronic effects of cannabis intoxication affect several biological systems, including the nervous (cognitive, psychiatric, psychomotor effects), respiratory, gastrointestinal, cardiovascular, immune, reproductive, and endocrine systems [18,56]. In the case of cannabis usage by mothers, there are also effects on fetal development. Further reports mention genotoxicity, mutagenicity, and oncogenesis [63]. The chronic toxicities associated with the inhalation and ingestion of cannabis products are as follows.

### 4.1. Inhalation

Chronic cannabis smokers are at an increased risk of developing bronchitis, emphysema, and squamous metaplasia, a pre-cancerous alteration in the tracheobronchial epithelium [56]. A previous study has revealed that cannabis users experienced chronic coughing, sputum, and wheeziness, as well as airway mucosal inflammation, goblet-cell and vascular hyperplasia, metaplasia, and cellular disruption. Furthermore, the inhalation of THC-containing products can also promote bronchoconstriction and the impairment of airway function [64]. A number of cannabis users have bullous emphysema. 

Besides this, cannabis smoking is a risk factor of cancer [65]. A meta-analysis revealed a significant association between cannabis smoking and an increased risk of lung cancer. Additionally, an elevated risk of testicular cancer was observed in those who have used cannabis for more than ten years [66]. Interestingly, marijuana usage is also associated with increased risk of oropharyngeal cancer, but a reduced risk of oral tongue cancer [67]. However, human papilloma virus (HPV) exposure may confound the association between marijuana use and oropharyngeal cancer [67]. An in vitro study reported that THC at concentrations equivalent to those detected in human serum after cannabis consumption could stimulate the proliferation of cancer cells [68]. Thus, the administration of cannabis to cancer patients should be used with caution. 

Cannabis smoke contains a combination of volatile chemicals, including ammonia, carbon monoxide, hydrocyanic acid, nitrosamines, and tar components (phenols, naphthalene, benzanthracene, and the pro-carcinogenic benzopyrenes) [15,69]. Several of these chemicals can inhibit the capacity of lung alveolar macrophages to kill bacteria in a bioassay system [66]. Importantly, a higher amount of tar was produced by cannabis cigarettes than tobacco cigarettes. Furthermore, cannabis-derived tar is more potent than cigarette-induced tar in inducing the expression of CYP1A1, an important enzyme catalyzing the metabolic activation of carcinogens [70]. In fact, the experiment showed that treatment with 3 ug/mL of marijuana-derived tar caused a greater increase in the expression of CYP1A1 mRNA in liver cells Hepa-1 than tobacco tar at the same dose [70]. 

Although cannabis smoke contains the same amount of carbon monoxide as cigarette smoke, the deep inhalation of cannabis smoke produced five times more carboxyhemoglobin than tobacco smoke. Such a high level of carboxyhemoglobin contributes to the development of atheromatous diseases (an occlusion or stenosis of a deep penetrating artery of the brain) [56]. Additionally, chronic cannabis smokers who also use tobacco have a greater prevalence of respiratory-symptom and histological alterations than those who smoke only cannabis [56].

### 4.2. Ingestion

Currently, there is limited clinical evidence of chronic toxicity related to the oral ingestion of cannabis products in humans. In animal studies, subchronic (90-day) oral toxicity tests of 9% hemp extract (of which 6.27% is CBD) and 91% olive oil have shown that the non-observable-adverse-event level (NOAEL) is 800 mg/kg bw/day for female and 400 mg/kg bw/day for male Sprague–Dawley rats [71]. The human equivalent acceptable daily dose for a male weighing 60 kg is around ((800/1000) × 60) = 48 mg. The toxic effects observed with doses above the NOAEL included an increase in liver weight, concomitant with the hypertrophy of liver cells and an increase in the adrenal-gland-to-body-weight ratio, along with the vacuolization of the adrenal zona fasciculate [71]. Another subchronic oral toxicity test showed that cannabis leaves at a dose of 172.9 mg/kg caused toxicities in the brain, liver, kidney, and testes in male albino rats [72]. Since cannabis-containing food products are gaining popularity, future clinical safety studies and risk assessments of chronic ingestion in humans are warranted. 

### 4.3. Exposure to Cannabis Regardless of Route

#### 4.3.1. Cognitive Performance, Psychomotor Performance, and Psychopathology

A range of undesirable health effects can occur following prolonged exposure to a substance, whether inhaled or ingested. THC binding to CB1 receptors can affect perception, cognition, motion, and related distress and psychotomimetic effects due to the selective adenylate cyclase activity suppression caused by CB1 activation [15]. In fact, a human study reported that the use of cannabis seven times per week or more (defined as chronic heavy use) was associated with deficiencies in mathematical skills, verbal expression, and memory recall [73]. Additionally, heavy cannabis users (smoked marijuana 22–30 days within 1 month) received lower scores on cognitive tests of attention and executive function than those of light users (who smoked marijuana 1–9 days within 1 month) [74]. The findings of this study and previous work suggest that the impact of cannabis use on cognitive impairment depends on the duration of usage [62,74]. Consistently, an in vitro study reported that the exposure of a hippocampus slice to 1–5 µL of a CB1 receptor agonist, WIN 55,212-2, impaired long-term potentiation (long-lasting increase in signal transmission between two neurons) and long-term depression (long-lasting decrease in synaptic strength). Since long-term potentiation and depression are crucial to learning and memory, the result suggests the adverse effects of cannabinoids on learning and memory [75]. 

Further central effects of cannabis include the disturbance of psychomotor performance, motor function, and response speed, which are believed to result from memory loss [15]. Furthermore, psychiatric symptoms such as ideation, delusions, and hallucinations can occur in people with prolonged cannabis use, and can continue after cessation [56]. Additionally, psychotic symptoms such as disorientation, visual and auditory hallucinations, paranoid ideations, mania, and schizophreniform psychosis were also found after chronic cannabis use [15,56,76]. Findings from recent studies suggest that the prevalence of these psychotic symptoms is associated with cannabis dependency in adolescents [76,77].

#### 4.3.2. Cardiovascular Systems

In addition to its effects on the central nervous system, cannabis has various physiological impacts, such as effects on cardiovascular, reproductive, digestive, and endocrine function, and oncogenesis effects. Cannabis exposure is documented to induce phasic systemic vasodilation, hypertension, tachycardia, high blood pressure, and elevated venous carboxyhemoglobin levels, as well as arrhythmias and myocardial infarctions [63,78]. A recent study estimated that lifetime myocardial infarction (ACI) could be raised by up to 8% among cannabis users [79]. A case report revealed that cannabis usage may be a potential risk for Takotsubo cardiomyopathy, characterized by transient left ventricular wall apical ballooning, which contributes to temporary left ventricular dysfunction [80,81]. Consistently, myocarditis related to cannabis usage was also reported in a 15-year-old teenager with no previous cardiac history [82]. Further research studies are warranted to investigate the risk posed by chronic cannabis use to cardiac function. 

#### 4.3.3. Hormone and Reproductive Systems

Several endocrine organs controlling sexual hormones as well as other hormones can be affected by cannabis exposure. The results of in vivo experiments suggest that THC suppresses the release of prolactin and growth hormone. In total, 0.4 and 4 µg of THC were microinjected into the third ventricle of rats, resulting in a reduction in plasma prolactin and growth hormone levels within 40–80 h. However, neither 5 × 10^−8^ nor 5 × 10^−9^ M of THC altered the release of these two hormones in anterior pituitary cell culture. The findings suggest that THC can influence the hormone release with a mechanism that is not directly related to the anterior pituitary [83]. In addition, cannabis has an anti-androgenic activity, and cannabinoids, such as THC, can bind to androgen receptors and operate on the hypothalamus–pituitary–adrenal (HPA) axis [15,56].

A systematic review provides the strongest evidence for cannabis-induced male fertility abnormalities. Cannabis reduces sperm count and concentration, induces abnormalities in sperm morphology, reduces sperm motility and viability, and inhibits sperm capacitation and fertilization [84]. Animal studies suggest the effects of cannabis manifest as testicular atrophy, lower libido, and decreased sexual function [84]. 

The effect of prolonged cannabis exposure on the reproductive system of female mice was studied by administering 6 and 12 mg/100 g body weight/day of an aqueous cannabis preparation to adult mice for 30 days [85]. Chronic cannabis exposure resulted in oxidative stress and damage in the reproductive system of mice, as indicated by a significant decrease in ovarian and uterine weight. Further studies found an increase in CB1 expression, and a decrease in total serum cholesterol, estradiol levels in circulation, 3β hydroxy steroid dehydrogenase (HSD), and 17β HSD enzyme activity in the ovarian tissues. This finding suggests that cannabis affects the female reproductive system by interfering with gonadal activity, steroidogenesis, and receptor expressions. In addition, cannabis can impair female fertility by blocking the hypothalamic release of gonadotropin-releasing hormone (GnRH), resulting in decreased estrogen and progesterone production, and leading to an ovulatory menstrual cycle [85]. According to data from animal research, cannabis usage can cause pubertal delay. In particular, THC is associated with a delay in sexual maturity and growth spurts. The systematic review’s findings demonstrate that the daily doses given to mice in the experiments ranged from 4 to 0.001 mg/kg, which are comparable to the doses of ∆9-THC exposure in humans at 0.1 to 3.9 mg/kg [86].

#### 4.3.4. Diseases Related to Long-Term Cannabis Use

Chronic cannabis abuse is linked with a clinical disorder recognized as cannabinoid hyperemesis syndrome (CHS) [87]. CHS is characterized by cyclic episodes of nausea and vomiting, abdominal pain, and frequent hot bathing. The precise mechanism of CHS is not well understood, and patients are frequently misdiagnosed for a long period of time [88,89]. Furthermore, it has been reported that exposure to cannabis is associated with a variety of cancer types [63], including lung cancer [90,91,92], head and neck cancer [93,94], laryngeal cancer [95], prostate cancer [96], cervical cancer [96], testicular cancer [97], and brain cancer [98]. The use of cannabis by pregnant women was also associated with an increased risk of neuroblastoma [99], nonlymphoblastic leukemia [100], and rhabdomyosarcoma [101] in their offspring. In conclusion, the chronic use of cannabis regardless of route can increase the risk of psychiatric disorders, heart diseases, abnormal fertility, and many types of cancer. However, the extent of adverse effects also depends on the individual variations in responses to cannabinoids. 

## 5. Individuality of Response

Uniquely, responses to cannabinoids follow a biphasic dose-dependence (hormesis) pattern, in which low doses and high doses could have opposite effects. Unfortunately, many factors can influence responses to cannabinoids. Therefore, the right dose for one person, with high efficacy and low toxicity, could be different for another. Thus, with the exact same dose and preparation, cannabis may be beneficial for some but may be toxic to others. In the following section, we describe the hormesis concept, and summarize factors influencing the individuality of responses, based on the existing literature. 

### 5.1. Biphasic Dose–Response

Individual responses to cannabinoids are explained by a biphasic dose response. Biphasic dose-dependence is also known as “hormesis”, characterized by low-dose stimulation and high-dose inhibition [102]. As shown in Figure 1, the key is to identify the hermetic zone in which the optimum doses provide maximum health benefits with minimum toxic effects. Biphasic dose response is a typical pharmacological phenomenon observed in several medications, hormones, and neurotransmitters [102,103]. Numerous reports demonstrate the biphasic effects of ∆9-THC and other cannabinoid agonists, in which low and high dosages commonly induce opposing effects [104,105,106]. THC showed opposite dose-related effects on cortical evoked responses in animal model studies. Low dosages raised the reaction amplitude, but greater doses reduced the effect [107]. 

In vitro and in vivo studies have demonstrated that cannabis has a biphasic effect on neurogenesis. Cannabinoids can increase neurogenesis at low doses. However, at higher concentrations, neurogenesis is impaired [106]. The effects of cannabinoids on acetylcholine release in the hippocampus were observed. Exogenously applied synthetic THC (WIN 55212-2) regulates septohippocampal cholinergic neurotransmission in a biphasic, dose-dependent manner. A low dose of synthetic THC stimulates hippocampal ACh efflux, while a high dose inhibits hippocampal ACh efflux [105]. Low dosages of ∆9-THC produce opposing effects on Sprague–Dawley rats compared to high doses. In total, 0.1 mg/kg ∆9-THC lowered intracranial self-stimulation (ICSS) thresholds and induced hyperactivity, while 1 mg/kg ∆9-THC raised ICSS thresholds and induced hypoactivity [108].

The involvement of CB1 receptors is implicated in the biphasic effects of cannabis on feeding behavior, motor activity, motivational processes, and anxiety-related reactions. The investigation on mice revealed that the anxiety-related effects of low doses of cannabis are mediated through the CB1 receptor on cortical glutamatergic terminals. Under high-dose conditions, the CB1 receptor on GABAergic terminals is required to elicit an anxiogenic-like effect [109]. Low dosages of ∆9-THC likely produce conditioned rewarding effects, whereas higher levels produce conditioned aversive effects [110]. Research on the endocannabinoid system’s control of food intake found that high dosages of cannabinoids showed orexigenic (appetite stimulant) effects, whereas low doses had anorexigenic (loss of appetite) effects [111]. 

Cannabinoids have a traditional inhibitory action on calcium channels. The compounds can act as agonists at high doses (micromolar range) and inhibit calcium entrance into cells, whereas significantly lower levels (nanomolar range) promote calcium entry [103]. Regarding the medicinal aspects, cannabinoids have long been used to treat nausea and vomiting; however, due to their biphasic effect, large dosages produce cannabinoid hyperemesis syndrome (CHS), characterized by repeated nausea and vomiting. The major contributor to the symptoms is THC compounds, which cause alterations in the endocannabinoid system by activating the cannabinoid 1 (CB1) receptor [104].

The challenge is that different people may have distinct optimum dosages of cannabinoids, and the toxic doses can be largely varied. Thus, the Lower-Risk Cannabis Use Guidelines (LRCUG) for reducing health harms state, as a general precaution, that there is no universally safe level of cannabis use [112]. The only reliable way to avoid any risk of harm from using cannabis is to abstain from its use [112].

### 5.2. Factors Affecting Individual Response to Cannabinoids

The hermetic zones of cannabinoids in which the optimum doses provide maximum health benefits with minimum toxic effects are determined by various factors, i.e., exposure factors, individual factors, and individual susceptibility factors. The details of each factor are summarized as follows. 

#### 5.2.1. Exposure Factors

The exposure factors, including route of exposure, frequency, duration, method of use, and interaction with other food, could influence pharmacokinetics (absorption and bioavailability, metabolism, distribution, and excretion). THC is highly lipophilic, so its bioavailability is high and can be influenced by other lipid-containing molecules. 

The route of exposure is the most important exposure factor. Inhalation is the fastest way to deliver cannabinoids to the body, as THC can reach its peak in plasma within just 3–10 min after inhalation [50,51]. Compared to inhalation, the ingestion of cannabis has more delayed kinetics (THC reaches peak in plasma within 1–2 h) as it takes a longer time to digest food and absorb cannabinoids into the blood circulation [49]. Moreover, the ingestion route yields a roughly 5–20% lower bioavailability of THC than smoking, as some is lost during digestion and absorption [49]. Since there are many harmful and carcinogenic compounds produced in cannabis smoke, the oral ingestion of cannabis transports much less hazardous chemicals than smoking [51,53]. Nevertheless, the effects of cannabinoids from the oral route may last longer than those from smoking, as it takes longer (up to 12 h) to return to the baseline when consuming orally, compared to 4 h via the inhalation route [50,54]. 

Besides the route, the delivery system is another important issue. Since the oral bioavailability of cannabinoids can be boosted by combining them with lipids, common delivery systems for cannabinoids include lipid/oil-based formulations and gelatin matrix pellets [113,114]. Thus, the effective and safe doses of cannabinoids for different formulations can also vary. Considering the consumption of cannabis in food products, the food matrix and interactions with other food are important subjects of concern. A previous study showed that food matrix, especially when fat-based, could increase the bioaccessibility of cannabinoids [115]. Furthermore, the absorption of cannabinoids can be increased if cannabis products are taken together with alcohol or high-fat/high-calorie meals [116,117]. Owing to the more complex pharmacokinetics, the consumption of cannabis-containing food products by oral ingestion may make it more difficult to predict the ideal dose of cannabinoids, compared to smoking.

#### 5.2.2. Individual Factors

Gender and age are important factors contributing not only to the physiologic function, but also the pattern, of cannabis use. Differences in sex influence the structures of genes, proteins, and cells, as well as tissue morphology and body function. Furthermore, the sexes are each associated with different norms, relationships, identities, and gendered structural environments. All of these aspects impact the patterns of cannabis use and responses [118]. 

In comparison to women, men use cannabis more frequently, in greater quantities, and with a greater variety of methods [118,119]. Interestingly, the responses to cannabis use in men and women are also different. During a “high” period after cannabis use, males are more likely to experience increased hunger, improved memory and enthusiasm, changes in time perception, and enhanced musical ability [119]. Women were more prone to hunger loss and a desire to clean (psychological effects, urges for repetitive behavior) [119]. 

Age is also a factor that influences sensitivity and response to cannabis exposure. Adolescence is not only a period of neurostructural and functional development, but is also a critical time in terms of behavioral and psychological vulnerabilities [120]. Evidence suggests that adolescence is a particularly vulnerable phase in relation to cannabis use, leading to adverse neurocognitive consequences that persist into adulthood. Early cannabis usage in teenagers is linked to reduced performance in cognitive tasks, as evidenced by poor educational outcomes, a higher school dropout rate, exasperated pre-existing characteristics, and a lower IQ [121]. Adolescent cannabis usage may raise the risk of psychosis, anxiety, and depression symptoms in late adolescence and early adulthood, which persist throughout maturity [122,123,124].

#### 5.2.3. Individual Susceptibility Factors

##### Genetic Polymorphism

Recent research has revealed that individual susceptibility to the toxicity of chemicals depends on genetic polymorphism. This affects the phenotypes of cannabis use, including the age of initiation, lifetime use, cannabis use disorder (CUD), and withdrawal and craving [125].

Several single-nucleotide polymorphisms (SNPs) of the cannabinoid receptor *CNR1* gene have been demonstrated to be linked with clinical diagnoses and phenotypes of cannabis use. The *CNR1* rs1049353 polymorphism is defined by a C > T substitution (a substitution of C by T) at location 88143916 on chromosome 6 [126]. According to a longitudinal study, the presence of the substituted T-allele was associated with a greater likelihood of cannabis use in teenage populations [127]. Besides the influence on cannabis use, polymorphism also affects the pattern of response. CC carriers of rs1049353 had increased satiety and decreased salience of cues following THC conditions, while T-allele carriers had considerably lower satiety [128]. An examination of the connection between rs1049353 variation and the response to cannabis smoking revealed that T-allele-carrying individuals, such as those with TC or TT, had a higher subjective response than CC individuals. The T-allele group showed a higher and more significant profile of mood states (POMS), including anger/hostility, depression/dejection, vigor, and elation [125]. According to case-control studies, the main C-allele of the rs1049353 polymorphism is related to cannabis dependency symptoms [129]. 

The rs2023239 of *CNR1* is defined by a T > C substitution (a substitution of T by C) at the 88150763 position of chromosome 6 [126]. The relationship between the SNP rs2023239 and cannabis withdrawal and craving was identified. Possessing one or more C-alleles increases the probability of developing increased degrees of withdrawal, negative effects, and craving more marijuana. C-carriers had a 20% greater marijuana dependency as a result of increased checklist scores and joint usage, which was increased by 30% per month compared to T/T groups [130,131]. Studying the POMS scale’s modulation following cannabis consumption and genotyping the rs2023239 polymorphic site indicates that minor C-allele carriers exhibit elevated anger/hostility, fatigue, tension/anxiety, and vigor scores over TT individuals [132]. The longitudinal assessment of adolescents among the non-Hispanic White population found that carriers of the minor allele (C-carriers) at rs806374 of the *CNR1* are more likely to use cannabis between the ages of 18 and 19, during the transition from high school to college [133]. 

Besides the cannabinoid receptor, genetic polymorphism of the *FAAH* gene-encoding *N*-acylethanolamine-hydrolyzing enzyme FAAH also affects cannabis response. The common *FAAH* SNP rs324420 has been implicated in withdrawal, craving, and drug cues [127]. This study indicates that the *FAAH* with C/C group experienced a more significant increase in cravings following abstinence than the FAAH with C/A group [133]. Cannabis users with the homozygous C-allele showed more severe withdrawal symptoms during abstinence and stronger cravings and cues following a trigger than those with the A-allele [134]. In addition, humans with the A/A genotype showed lower FAAH expression and activity. C/C or C/A genotype populations were more likely to be THC-dependent than A/A genotype populations [135]. 

The *CHRNA2* gene encodes the nicotinic acetylcholine receptor (nAChR) α2subunits [125,136]. Polymorphisms in *CHRNA2* were found to be strongly associated with nicotine-dependence (ND) symptoms, mediated by nicotinic acetylcholine in European and African-American family populations [137,138]. The genome-wide association study (GWAS) based on 51,345 cannabis-using and nonusing individuals across two cohorts showed that SNP rs56372821 of *CHRNA2* was more prevalent in people with cannabis use disorder [139]. 

*ATP2C2* is a gene on chromosome 16 that encodes calcium-transporting ATPase. *ATP2C2* has been significantly related to the age at which cannabis use begins, which has been linked to various maladaptive behaviors. A GWAS meta-analysis was conducted in 24,953 individuals from nine European cohorts, Australia, and the United States. The rs1574587 intronic variant of *ATP2C2* exhibited the highest association with age at first cannabis use [140]. 

CADM2 is a synaptic cell-adhesion molecule (SynCAM family) in the immunoglobulin (Ig) superfamily [141]. It has been connected with alcohol consumption, reproductive success, and several personality traits, including a risk-taking, optimistic, and carefree character [142,143]. Three *CADM2* polymorphisms, rs2875907, rs1448602, and rs7651996, have been linked with lifetime cannabis usage, according to a GWAS analysis of 184,765 individuals [143]. In addition, the existence of rs2875907 and rs7651996 is likely to increase cannabis use over a lifetime, but rs1448602 is associated with a decreased likelihood of use [125,143]. 

NCAM1, or neural cell-adhesion molecule 1, is a member of the immunoglobulin superfamily and is associated with neurogenesis and dopaminergic neurotransmission [143]. The first meta-analysis of GWAS and gene-based tests on 32,330 individuals from 13 cohorts revealed the association between *NCAM1* polymorphism and lifetime cannabis usage [141]. Additional research has indicated that the SNP rs9919557 of *NCAM1* is associated with lifetime cannabis use [143]. Further research on 1284 individuals from the Mexican-American population also found an association between rare SNPs upstream of *NCAM1* rs7932341 and lifetime cannabis usage and depression [144].

Genetic polymorphisms in THC-metabolizing enzymes also influence cannabis toxicity. For example, compared to wild-type alleles, cannabis users who carry genetic variants in cytochrome P450 2C9 (*CYP2C9*3*) have shown decreased enzyme activity leading to lower concentrations of the psycho-inactive Δ9-THC-COOH and a higher ratio of Δ9-THC/Δ9-THC-COOH [144]. Thus, people with such genetic polymorphisms would have a longer half-life of THC, which increases its bioavailability and, presumably, its chance of becoming toxic [145]. *CYP2C9* polymorphisms are highly prevalent in certain racial groups, especially Caucasians [146]. 

Since CYP2C9 and CYP3A4 are used in metabolizing not only THC but also a variety of drugs, interactions between cannabis use and other drugs are commonly observed [145]. For example, the co-administration of Sativex Oromucosal Spray (extracts of Cannabis leaf and flower, containing 27 mg of ∆9-THC and 25 mg of cannabidiol) and ketoconazole (400 mg; 5 days) increased the bioavailability of THC by 27% and 11-OH-THC by 204% [147]. The reason is that ketoconazole is a strong CYP 3A4-inhibitor [139]. In the study, 100% of the participants experienced an adverse event, primarily in the central nervous system [147]. In contrast, the co-administration of a cannabis product with a strong CYP3A4- and CYP2C19-inducer, such as rifampicin (600 mg; 10 days), resulted in a decrease in the maximum plasma concentration (Cmax) of THC by 36% and 11-OH-THC by 87% [147]. 

##### Epigenetic Regulation

Besides the genetic polymorphism affecting the structure of the gene, epigenetic regulatory mechanisms including DNA methylation, histone modification, and noncoding RNAs (ncRNAs) pathways also affect cannabis response [148,149]. 

DNA methylation has been observed on CpG sites within the *MARC2* and *CUX1* genes of the cannabis exposure group. MARC2 is involved in the physiologic response to antipsychotic medication in schizophrenic patients, while CUX1 plays a role in neural development [150]. CB1 receptor expression is also affected by DNA methylation. The study shows that CB1 receptor expression is enhanced in the peripheral blood lymphocytes of schizophrenia patients who use cannabis. The expression is inversely associated with *CNR1*-promoter methylation [151].

Catechol-O-methyltransferase (COMT) is an enzyme that degrades dopamine. The Val108/158Met polymorphism of the *COMT* gene affects the COMT enzyme activity, and consequently alters dopamine concentrations. A study of human adolescent peripheral blood cells indicated that the methylation of the *COMT* gene is related to cannabis use among adolescents. Adolescents with the Met/Met genotype and high rates of *COMT* promoter methylation were less likely to be heavy cannabis consumers than those with the Val/Val or Val/Met genotype [149,152].

THC treatment significantly alters the methylation and acetylation of histones. Four histone methylations and one histone acetylation (H3K4me3, H3K9me3, H3K27me3, and H3K36me3) were altered following THC exposure in the lymph node cells of developing mice. These modifications also affected the expressions of gene promoters and dysregulated the downstream genes and noncoding RNA [153]. A study in an adult rat brain (nucleus accumbens (NAc)) with parenteral THC exposure revealed an alteration in the histone modification profile of the *Drd2* gene, which encodes dopamine receptor D2. The data indicate decreased levels of H3K4me3, increased levels of H3K9me2, and reduced amounts of *Drd2* gene mRNA [154]. More evidence was uncovered in a study of nucleus accumbens shell (NAcsh) in adult rats exposed to THC. THC exposure increases the expression of *Penk*, the gene encoding the opioid neuropeptide enkephalin involved in reward-related behaviors. THC also promotes *Penk* upregulation by decreasing H3K9 methylation in the NAcsh, interrupting the regular developmental pattern of H3K9 methylation [155].

Alteration in miRNA profiles has also been linked to THC exposure. The miRNA expression profile of THC-exposed SIV-infected macaques was performed using intestinal pinch biopsies. Compared to the control group, the THC exposure group had increased expressions of miR-10a, miR-24, miR-99b, miR-145, miR-149, and miR-187. Furthermore, miR-99b has been shown to upregulate the target gene, NOX4, affecting the composition of intestinal epithelial cells [156]. Another study demonstrated that THC-treated mouse myeloid-derived suppressor cells (MDSCs) display distinct miRNA expression, with the significant overexpression of miRNA-690 [157].

## 6. Evidence-Based Clinical Safety and Efficacy of Cannabis Products

Although the use of cannabis and cannabinoids has been legalized in several countries, knowledge regarding the efficacy and safety of cannabis and cannabinoids in general populations is still limited. Recent studies have shown that distinct developmental stage, pathophysiology, age, absorption, and metabolism result in variations in the safety and efficacy of cannabis use in pediatric and adult patients [158]. A 2017 systematic review of cannabis as a medicinal treatment for children and adolescents revealed its effectiveness against chemotherapy-induced nausea, vomiting, and epilepsy. However, the data were limited, and more research is required to determine the significance of medical cannabis in young people [159]. 

A systemic review and meta-analysis published in 2021 revealed that CBD can effectively treat epilepsy in a pediatric population. However, cannabidiol is related to appetite suppression and impaired physical development. Additionally, adverse mental events have been observed with CBD and THC analog treatments [160]. Further clinical trials with long-term follow-up are necessary to identify the chronic adverse effects of cannabis use. Due to changes in drug kinetics associated with aging and increased risks of adverse drug reactions, the efficacy and safety data obtained from studies in adult populations cannot be applied to older populations [161]. 

Aging is associated with a cumulative decrease in several physiological functions, resulting in losing the ability to maintain homeostasis [162]. A systematic review of medical cannabinoids use has revealed that the information about the use of cannabis in elderly patients is still limited, and the data are insufficient to provide any recommendations [163]. According to the paper, sedation-like symptoms were frequently observed in older subjects during cannabis treatment [163]. Likewise, another systematic review and meta-analysis found that the usage of natural and synthetic cannabinoids in people over the age of 50 is associated with increased side effects. Therefore, THC:CBD mixtures may be less acceptable for use in individuals over the ages of 65 or 75 [164]. 

The pros and cons of cannabis use have been summarized in several articles [165,166,167]. The pros of cannabis use include reducing inflammation, relieving pain, decreasing nausea, alleviating depression and anxiety, and improving sleep quality. Thus, it has been legalized for medical, dietary, and recreational use in several countries around the world. For people with chronic brain disorders, cannabis use reduces some neurological and psychiatric symptoms, i.e., alleviating pain and spasticity in people with multiple sclerosis, reducing tremors, rigidity, and pain in people with Parkinson’s disease, and improving the quality of life of patients with amyotrophic lateral sclerosis by improving appetite and decreasing pain and spasticity [165,166,167]. Nevertheless, the cons of chronic cannabis use include memory and cognitive impairments, dependence, and worse psychiatric symptoms in people with schizophrenia and bipolar disorder [165,166,167]. Therefore, cannabis use is still illegal and considered a social stigma in many countries. 

## 7. Challenges in Establishing Safe Doses of Dietary Cannabis Products

Several countries around the world have legalized cannabis products for medical, recreational, and dietary purposes [1]. However, most countries only regulate the amount of Δ9-THC in cannabis products, with no specified safe dose for consumption. The allowed amount of Δ9-THC in cannabis oil from cannabis seeds and the total content in food in European countries, as well as Australia and New Zealand, were 5–20, 2–10, and 0.02–20 mg/kg of final products [168]. Thai law allows no more than 1.6 mg of Δ9-THC and no more than 1.41 mg CBD per pack of food product containing crude cannabis or hemp. The amount must be quantified by high-performance liquid chromatography or better techniques [169]. For cannabis extracts used in food products, Thai regulation permits no more than 0.2% of THC by weight and no less than 30% of CBD [170]. Furthermore, Thai law prohibits the use of cannabis, hemp, and extracts in caffeine-containing drinks, food, and dietary supplements for babies and infants [171].

In 2015, the European Food Safety Authority (EFSA) established an acute reference dose (ARfD) of 1 μg Δ9-THC/kg body weight (bw). This safe limit is for one-time consumption as a single dose. The dose was calculated from the consumption of milk and dairy products, resulting from the use of hemp-seed-derived feed materials [172]. However, in 2020, EFSA conducted an exposure assessment of Δ9-THC from cannabis products among EU countries, and reported that the actual consumption mostly exceeded the reference dose of 1 μg/kg bw [170]. 

Acute exposure in adult consumers was estimated based on the highest reliable percentile of occurrence, i.e., hemp seeds (up to 9 μg/kg bw), hemp oil (up to 21 μg/kg bw), tea (infusion) (up to 208 μg/kg bw), breakfast cereals (up to 1.3 μg/kg bw), pasta (raw) (up to 6.4 μg/kg bw), bread and rolls (up to 1.3 μg/kg bw), bread and rolls from hemp flour (up to 4.1 μg/kg bw), cereal bars (up to 0.3 μg/kg bw), fine bakery wares (up to 5.1 μg/kg bw), chocolate (cocoa) products (up to 1.1 μg/kg bw), energy drinks (up to 0.2 μg/kg bw), dietary supplements (up to 9.9 μg/kg bw), and beer and beer-like beverages (up to 41 μg/kg bw) [173]. These facts will likely lead to more careful thought being applied in setting up a safe dose for cannabis products. Besides EFSA, Beitzke and Pate have proposed a tolerable upper intake level of Δ9-THC of 0.5 mg/day or 7 ug/kg bw, which was calculated based on both the reference dose for THC traces in foods and the pharmacokinetic data of THC in humans [174]. 

In 2022, EFSA paused their considerations aiming to establish a safe dose of CBD. The chair of EFSA’s expert Panel on Nutrition, Novel Foods, and Food Allergens (NDA) commented that the existing research data are inadequate to assess the safety and establish a safe dose for CBD. There is insufficient information regarding the effect of CBD on the liver, gastrointestinal tract, endocrine system, nervous system, and psychological well-being. Since animal studies have shown adverse effects of CBD exposure on the reproductive system, it is crucial to determine if these effects are also seen in humans [175]. Due to the lack of adequate toxicological information, so far, no countries have set a safe level of Δ9-THC or CBD in cannabis products continuously consumed as food, otherwise known as acceptable daily intake (ADI). 

Challenges in defining the safe dose of Δ9-THC or CBD in cannabis products are derived from the individuality of responses, as described in previous sections of this review. Thus, one specific dose may be beneficial for some, but could be harmful to others. Furthermore, for dietary use, it is worth noting that the amount of THC released from food varies, since it also depends on the cooking process. Raw cannabis contains Δ9-tetrahydrocannabinolic acid A (THCA-A), a nontoxic substance that can be decarboxylated into the intoxicating substance THC, and cannabidiolic acid (CBDA), which could be decarboxylated into CBD [176]. Exposure to light or cooking with heat can activate the decarboxylation process, resulting in the generation of THC and CBD. Both heating temperature and duration affect the amount of THC and CBD released [177]. Currently, laws for cannabis products around the world mostly regulate the amount of Δ9-THC, but not Δ9-THCA-A [168]. Since the conversion of THCA-A and CBDA to THC and CBD can be varied, it is difficult to predict the actual amount of THC and CBD that humans really consume. Consequently, it is challenging to estimate the health risks associated with cannabis consumption through food. 

In medical practice, it is advised to use no more than 30 mg of THC, and this must be in combination with CBD so as to counteract the adverse effects. Furthermore, due to the vast variation in the individuality of responses, there are no definite recommended doses of THC or CBD for medical treatment. It is recommended to “start low, go slow, and stay low”, which means starting with a low dose of cannabinoids and gradually increasing the dose until reaching the lowest dose that provides the desired efficacy, then maintaining the dose. For dietary use, the current evidence is still insufficient to make any solid recommendations. Taking into consideration the individuality of responses, vulnerable subjects such as children and the immunocompromised elderly should avoid consuming cannabinoid-containing food products. For healthy adults, it is better to use the same concept as in medical practice, which is to “start low, go slow, and stay low”.

Besides the dose, the method of delivery is another crucial point of concern. Owing to its low solubility, rapid metabolism, poor bioavailability, and uncertain pharmacokinetics, the efficacy of cannabinoid is erratic. To improve the efficacy of cannabis use, new delivery systems via oral, transdermal, pulmonary, and transmucosal routes have emerged, as summarized in recent review articles [116,117]. Examples of delivery systems include encapsulation within micelles, ester prodrugs, gel formulations, and buccal and sublingual formulations [116,117]. 

Many delivery systems are based on the concept that the oral bioavailability of cannabinoids can be boosted by combining them with lipids. Lipid-/oil-based formulations and gelatin matrix pellets are two commonly used and effective formulations. Recently, new cannabinoid delivery systems have emerged. Examples include nanotechnology such as lipid-based nanosized and self-emulsifying delivery, and polymeric nanoformulations [116,117]. These new approaches improve dissolution, stability, and bioavailability. However, it is still unclear if these novel formulations will affect the safety profiles of cannabis products. More toxicological studies are warranted to clarify this. 

Since the rate and degree of absorption can be increased when taking cannabis products with alcohol or high-fat/high-calorie meals [119,120], establishing a safe dose of cannabis use for dietary purposes must also take the delivery system and pharmacokinetic interactions with other foods into consideration. 

## 8. Conclusions and Future Perspectives

### 8.1. Conclusions

This review highlights individual variations in responses to cannabinoids, which lead to the challenge of establishing recommended effective doses and standard safe doses of cannabis products for the general population. As shown in Figure 2, the factors affecting responses to cannabis products include exposure factors, individual factors, and individual susceptibility factors. 

Regarding the exposure factors, inhalation usually carries more toxic side effects than ingestion. The chronic smoking of cannabis, especially with deep inhalation, can increase the risk of lung, oropharyngeal, and testicular cancer. High-dose consumption could lead to acute toxicity. Chronic use is linked with reduced fertility in both males and females, as well as cannabinoid hyperemesis syndrome (CHS), which refers to cyclic episodes of nausea and vomiting, abdominal pain, and frequent hot bathing. 

Regarding the individual factors, males are more likely to have increased hunger, improved memory and enthusiasm, and enhanced musical ability when high. Women are more prone to experience hunger loss and a desire to clean. Adolescence is a vulnerable phase in terms of cannabis use, as it can lead to adverse neurocognitive consequences that persist into adulthood. 

Regarding the individual susceptibility factors, genetic polymorphisms of the *CNR1*, *FAAH*, *CHRNA2*, *ATP2C2*, *NCAM1*, and *CADM2* genes can influence both the likelihood of the use of and addiction to cannabis, and the susceptibility to cannabis toxicity. Individual susceptibility factors include genetic polymorphism and epigenetic regulation. The polymorphism of gene-encoding metabolizing enzymes such as CYP2C9 and CYP3A4 can result in the increased bioavailability of THC and susceptibility to adverse effects. The toxicity of THC could be amplified or inhibited if cannabis products are consumed together with other drugs that can modulate these enzymes. Epigenetic factors such as DNA methylation can influence cannabis use. For example, methylation of the *COMT* gene is related to cannabis use among adolescents. Reciprocally, the THC in cannabis significantly alters histone acetylation and microRNA (miRNA) profiles, resulting in alterations in gene expression. 

Owing to the individuality of responses, it is difficult to establish a safe dose of cannabinoids for the general population. Therefore, for medical, recreational, or dietary purposes, one should start low, go slow, and stay low. For regulatory purposes, the setting of a safe dose of cannabinoids for the continuous consumption of cannabis-containing food products should take into consideration both the toxicological data and the factors affecting the individuality of response, as well as the delivery system and the pharmacokinetic interactions with other foods. The calculation of the acceptable daily intake (ADI) level should incorporate more uncertainty factors in order to subtract the non-observable adverse event level (NOAEL) that is obtained from the toxicological data.

### 8.2. Future Questions to Be Addressed

Due to several variables, the intensity of each individual’s reaction to cannabis and the types of these reactions are entirely unique. Therefore, it is challenging to determine safe or effective dosages for the general population. To accurately predict the response and optimum doses for each individual with efficacy and safety, more research studies should be undertaken by means of metabolomic profiling after THC exposure, and genomic profiling of responders and nonresponders at various doses. These insights can be used to set up a predictive equation model that may be useful for medical, recreational, and dietary purposes. 

For toxicological studies, one important area that still requires more research is chronic toxicity from ingestion. Since the legalization of cannabis products in several countries, various forms of cannabis-containing food products have become available on the market. Unfortunately, the existing evidence is mostly present in the form of case reports. More research on humans regarding the safety of chronic oral ingestion is needed. Further studies are also required to advise on safe usage in specific groups, especially in elderly people, who may benefit from the pain-relieving and antidepressive effects of cannabinoids. 

To precisely establish the regulated concentration of cannabinoids in food products, risk assessments for the human consumption of cannabis-containing food products are warranted to understand the whole picture and raise awareness of the potential hazards of cannabis exposure and overdosing. In the meantime, developing rapid, reliable, and cost-effective techniques for monitoring the number of active substances in products is crucial for facilitating cannabis-related research. Furthermore, novel strategies to reduce the toxicity of cannabinoids in cannabis products are an important area of research, especially for high-risk groups such as children, pregnant woman, and people with genetic or epigenetic susceptibility. 

## Figures and Tables

**Figure 1 molecules-28-02791-f001:**
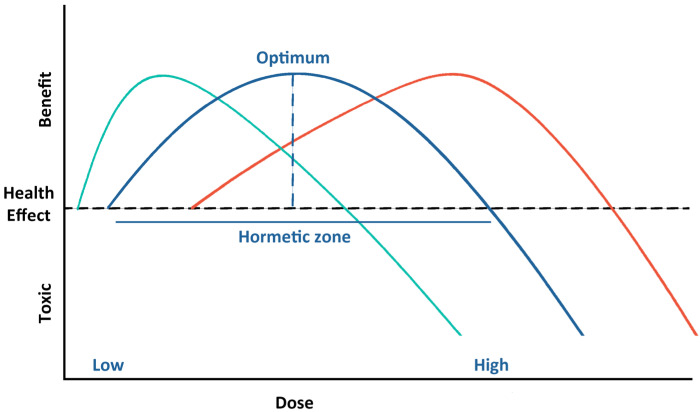
The biphasic-dose-response concept of cannabinoids’ effects on an individual. The blue line represents the theoretical dose-response curve between dose and health effects regarding benefits and toxicity. The vertical dash line indicates the optimum dose to provide health benefits. The range of health-beneficial doses is presented in the hermetic zone. The green and red lines represent the trends in sensitive and tolerant populations, respectively.

**Figure 2 molecules-28-02791-f002:**
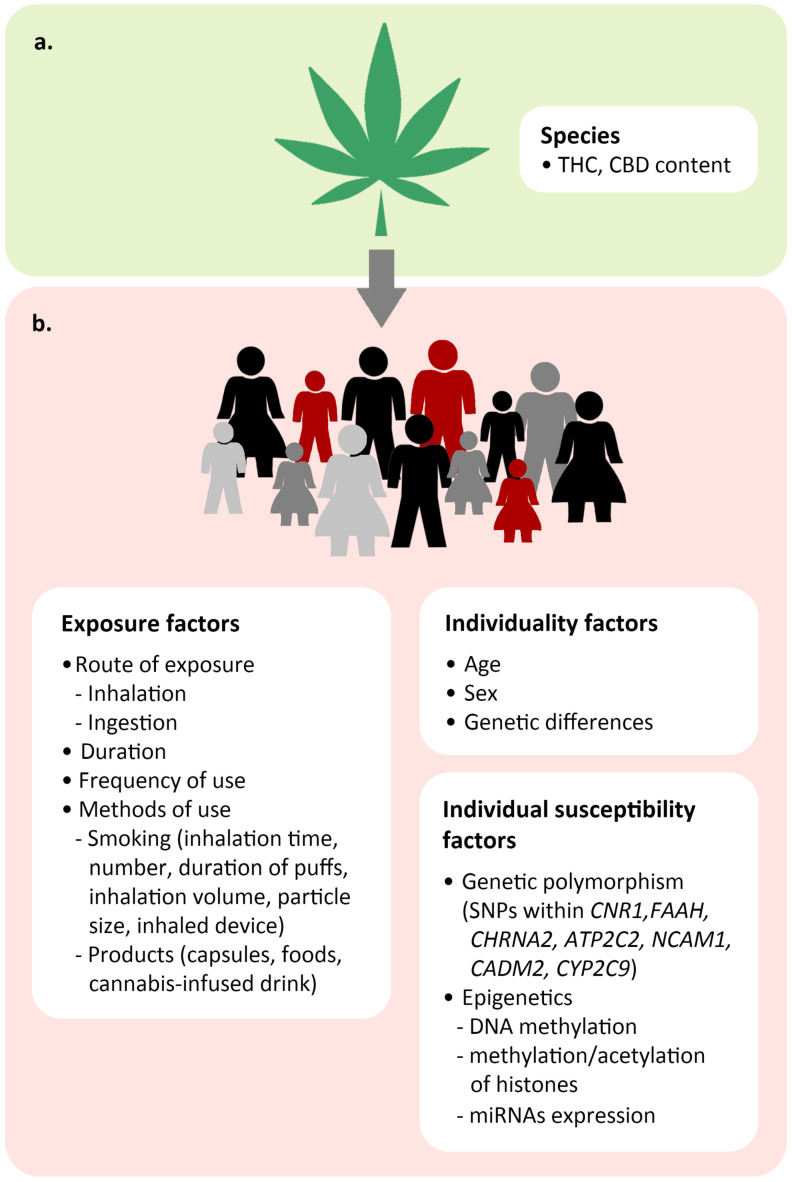
Factors influencing acute and chronic cannabis toxicity in an individual. (**a**) Depending on the species, the amounts of CBD and the psychoactive component THC can vary. (**b**) The toxicity of cannabis depends on exposure factors (route of exposure, duration, frequency, and method of use), individuality factors (age, sex, and genetic differences), and individual susceptibility factors (genetic polymorphism and epigenetics).

## Data Availability

Not applicable.

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
