# Peer review of "An Individuality of Response to Cannabinoids: Challenges in Safety and Efficacy of Cannabis Products"

_molecules, 2023, doi:10.3390/molecules28062791_

Round 1
Reviewer 1 Report
1. A comprehensive table is needed for the used drug with their chemistry and pharmacology data.
2. Pros and cons of cannabis must be added.
3. its therapeutic use as delivery system must be added.
4. Geographical data of this plant and chemistry part must be updated.
Author Response
- A compressive table is needed for the used drug with their chemistry and pharmacology data.
Response: Thank you for your suggestion. As we review the literature, we found several other recent review articles that extensively summarized this point. Therefore, we briefly describe the concept on page 2 lines 58-66, and provide a list of references [11-13] for further reading.
- Pros and cons of cannabis must be added.
Response: Thank you for your suggestion. To address this point, on page 14, lines 659-670, the pros and cons of cannabis use have been added with three recent review articles [165-167] as references for further reading.
- Its therapeutics used and delivery system must be added.
Response: Thank you for your suggestion. Therapeutic use has been reviewed in other literature so we added a reference for further reading on the first paragraph of page 2. For delivery systems, on page 10, lines 473-477, and page 15 lines 737-750, delivery systems of cannabis use have been added with multiple references [116-117] for further reading.
- Geographic data of this plant and chemistry part must be updated
Response: Thank you for your suggestion. Geographical origins of the cannabis plant and variations in chemical components have been added with references [18], [27-29] on page 2 lines 78-81, and page 3 lines105-117.
Reviewer 2 Report
The title of Manuscript should be reframed
Uniform same font in overall MS
Professional English editing is needed to correct some errors .
Whole MS is in plagiarism
Reduce the % of plagiarism upto 10%
Add one Table of bioactive compounds
All genus and species name should be be in Italics
Line 52-55: Rewrite
Maximum reference cited in MS is added after full stop. Like line 70 after fatty acid full stop will be there. Correct it in whole MS
Line 99-100: Rewrite
Line 179-191 rewrite contains plagiarism
Line 596-619: Rewrite contains lots of plagiarism
Line 209 remove full stop after care unit
Rewrite line 602-609
Line 634 correct no to not
Line 618 space between or7
Line 588 correct no to not
The writing could be improved by strengthening the connectivity between paragraphs. There are several places where new topics are introduced and connections to the previous subject are not clear. Read whole manuscript and correct wherever required.
Introduction:
The introduction does not clearly state the purpose of the research – please amend.
Conclusions
The conclusions are too general, format according to future aspects. Please make them more specific.
Carefully read whole manuscript line by line and improve the sentence formation
Cross check all references and style of reference according to Journal format, use abbreviation of journal name in reference
Author Response
- The title of the Manuscript should be reframed
Response: Thank you for your comment. This review aims to highlight the concept of individual variation in response to cannabinoids, which leads to the challenge of establishing standard safe doses of cannabis products for the general population. Therefore, we think the title “An Individuality of Response to Cannabinoids: Challenges in Safety and Efficacy of Cannabis products" is most fitted with the aim. Other reviewers do not have any comments on the title. So we prefer to keep the original title.
- Uniform same font over MS
Response: Thank you for your suggestion. We have corrected it throughout the document.
- Professional English editing is needed to correct some errors.
Response: The manuscript has been professionally edited as suggested.
- Reduce the % of plagiarism up to 10%
Response: After revision according to the reviewers’ suggestion, all parts of the manuscript have been checked for plagiarism via https://www.duplichecker.com/ and found 0% plagiarism.
- Add one Table of bioactive compounds
Response: The list of bioactive compounds in cannabis and their structures has been reviewed in the previously published literature. So we added the references [21] for suggested reading as shown on page 2, lines 87-88.
- All genus and species names should be in Italics
Response: Thank you for your suggestion. We have corrected it throughout the document.
- Line 52-55: Rewrite
Response: Thank you for your suggestion. We revised the sentences to “Furthermore, defining the safe dose of cannabis products based on a specific active compound such as THC or cannabidiol (CBD) is insufficient for predicting the effects. Due to the differential profile of the other 140 phytocannabinoid contents, a wide range of effects can be observed."
The statement now appears on page 2, lines 52-55
- Maximum reference cited in MS is added after full stop. Like line 70 after fatty acid full stop will be there. Correct in whole MS
Response: Thank you for your suggestion. We have corrected it throughout the document.
- Line 99-100: Rewrite
Response: Thank you for your suggestion. We revised the sentences to " AEA is hydrolyzed to arachidonic acid and ethanolamine by 129 fatty acid amide hydrolase (FAAH), while 2-AG is hydrolyzed to arachidonic acid and 130 glycerol by monoacyl-glycerol lipase (MAGL)". The statement now appears on page 3 lines 129-131.
- Line 179-191: rewrite contain plagiarism
Response: Thank you for your suggestion. We revised the sentences to“About 50% of the THC and other cannabinoids in cannabis are released into the 211 smoke. Experienced users often inhale deeply and hold the smoke in the airways for a few 212 seconds before exhaling it.” The statement now appears on page 5 lines 211-213.
- Line 596-619: rewrite contain plagiarism
Response: Thank you for your suggestion. We revised the sentences and checked for plagiarism via https://www.duplichecker.com/ and found 0% plagiarism.
- Line 209 remove stop after care unit.
Response: Thank you for your suggestion. We have corrected it. The statement now appears on page 5 line 241.
- Rewrite lines 602-609
Response: Thank you for your suggestion. We revised the sentence and moved it to the end of the paragraph for better clarification appears on page 15 lines 707-710.
- Line 634 correct no to not
- Line 618 space between or7
- Line 588 no to not
Response: Thank you for your suggestion. We have corrected
- The writing could be improved by strengthening the connectivity between paragraphs. There are several places where new topics are introduced and connections to previous subjects are not clear. Read whole manuscript and correct wherever required
Response: Thank you for your suggestion. We have revised the manuscript as suggested. We have rewritten the line, corrected the typo and mistake, and extensively revised the writing as suggested.
- Introduction: The introduction does not clearly state the purpose of the research-please amend.
Response: Thank you for your suggestion. We rewrite the last paragraph of the introduction on page 2 lines 66-70 to specify the gap of knowledge and the purpose of our review.
- Conclusion: The conclusions are too general, format according to future aspects. Please make more specifics.
Response: Thank you for your suggestion. We rewrite the conclusion to be more specific as suggested.
- Carefully read whole manuscript line by line and improve the sentence formation.
Response: Thank you for your suggestion. We rechecked the whole manuscript and corrected the structure and grammatical errors.
- Cross check all reference and style of reference according to Journal format, use abbreviation of journal name in reference.
Response: Thank you for your suggestion. We checked and edited the references throughout the document.
Reviewer 3 Report
The review "An Individuality of Response to Cannabinoids: Challenges in Safety and Efficacy of Cannabis products" by Sarunya Kitdumrongthum and Dunyaporn Trachootham provides up-to-date insight into the mechanism of actions, acute and chronic toxicities from inhalation and ingestion, factors affecting toxicological response to cannabis products and the individual variations, and evidence-based clinical safety and efficacy of cannabis products. The novelty of this review should be carefully described, and I would suggest the authors polish the language and improve the logicality. My main concerns are listed as follows:
(1) Abstract: This section should highlight the main findings of the current review, instead of background.
(2) Line 88, here should be written as THC [21]. instead of THC. [21]. Please check the similar mistakes, as it appears many times for references.
(3) Line 399-403: Where comes the data in Figure 1? This should be seriously treated. Please check them in other text expression in the whole text.
Author Response
The review "An Individuality of Response to Cannabinoids: Challenges in Safety and Efficacy of Cannabis products" by Sarunya Kitdumronghtum and Dunyaporn Trachootham provide up-to-date insight into the mechanism of actions, acute and chronic toxicities from inhalation and ingestion, factors affecting toxicological response to cannabis products and the individual variations, and evidence-based clinical safety and efficacy of cannabis products. The novelty of this review should careful described, and I would suggest the authors polish the language and improve the logicality.
Response: Thank you for your suggestion. We added some statement in the last paragraph of introduction to describe the gap of knowledge and the purpose of this review. The novelty of this review is now clearer. Also, we revised the manuscript to improve the logical flow as suggested.
- Abstract: This section should highlight the main finding of the current review, instead of the background.
Response: Thank you for your suggestion. We revised the abstract as suggested.
- Line 88, here should be written as THC [21]. Instead of THC.[21]. Please check the similar mistakes, as it appears many times for references.
Response: Thank you for your suggestion. We rechecked and corrected the whole manuscript.
- Line 399-403: Where comes the data in Figure 1? This should be seriously treated. Please check them in other text expression in the whole text.
Response: Thank you for your comment. The illustration in Figure 1 is our proposed concept of individual response to cannabinoids which has been summarized from the review articles and research studies.
- The concept of biphasic dose-dependence, also known as "hormesis," characterized by a low dose stimulation and a high dose inhibition, was explained in the review by Calabrese, E.J., 2013
Reference: Calabrese, E.J. Hormetic mechanisms. Critical Reviews in Toxicology 2013, 43, 580-606, doi:10.3109/10408444.2013.808172.
- The cannabinoid duality effects (opposing effects), which is the optimum doses provide maximum health benefits with minimum toxic effects, was explained in the review by Sarne, Y.,2019
Reference: Sarne, Y. Beneficial and deleterious effects of cannabinoids in the brain: the case of ultra-low dose THC. The American Journal of Drug and Alcohol Abuse 2019, 45, 551-562, doi:10.1080/00952990.2019.1578366.
A brief description of the illustration was included in the manuscript. (Page 9, Line 425-429)
The supportive study (ref. 102-108) was included in the manuscript. (Page 9, Line 407-423)
Round 2
Reviewer 1 Report
Accept
Author Response
Thank you.
Reviewer 3 Report
I would suggest the authors do the revisions in a careful and detailed way.
Author Response
- I would suggest the authors do the revisions in a careful and detailed way.
- Extensive editing of English language and style required
Response: Thank you for your suggestion. We checked the manuscript thoroughly and revised it carefully as suggested. The manuscript is professionally edited by the MDPI language editing system. The certificate of approval is attached.
